# Defying Gravity to Enhance Power Output and Conversion Efficiency in a Vertically Oriented Four-Electrode Microfluidic Microbial Fuel Cell

**DOI:** 10.3390/mi15080961

**Published:** 2024-07-27

**Authors:** Linlin Liu, Haleh Baghernavehsi, Jesse Greener

**Affiliations:** 1Département de Chimie, Université Laval, Québec, QC G1V 0A6, Canada; 2CHU de Québec, Centre de Recherche du CHU de Québec, Université Laval, Québec, QC G1L 3L5, Canada

**Keywords:** microfluidics, microbial fuel cells, electrogenic bacteria, *Geobacter sulfurreducens*, bioelectrochemical systems, bioelectrochemistry, power density, conversion efficiency

## Abstract

High power output and high conversion efficiency are crucial parameters for microbial fuel cells (MFCs). In our previous work, we worked with microfluidic MFCs to study fundamentals related to the power density of the MFCs, but nutrient consumption was limited to one side of the microchannel (the electrode layer) due to diffusion limitations. In this work, long-term experiments were conducted on a new four-electrode microfluidic MFC design, which grew *Geobacter sulfurreducens* biofilms on upward- and downward-facing electrodes in the microchannel. To our knowledge, this is the first study comparing electroactive biofilm (EAB) growth experiencing the influence of opposing gravitational fields. It was discovered that inoculation and growth of the EAB did not proceed as fast at the downward-facing anode, which we hypothesize to be due to gravity effects that negatively impacted bacterial settling on that surface. Rotating the device during the growth phase resulted in uniform and strong outputs from both sides, yielding individual power densities of 4.03 and 4.13 W m^−2^, which increased to nearly double when the top- and bottom-side electrodes were operated in parallel as a single four-electrode MFC. Similarly, acetate consumption could be doubled with the four electrodes operated in parallel.

## 1. Introduction

Bioelectrochemical systems (BESs), which combine microbiology and electrochemistry, are playing an increasingly important role in the sustainable technology sphere, with applications including wastewater treatment [1,2,3,4,5,6,7,8], electricity production [9,10,11,12], environmental remediation [13,14,15], chemical synthesis [16,17], and biosensors [18,19]. Common BESs include microbial fuel cells (MFCs) [20,21,22] and microbial electrolysis cells (MECs) [23,24]. In MFCs, chemical energy is converted into molecular products, such as hydrogen and methane, and electrons [16]. As with other BESs, electrogenic bacteria are indispensable. These usually form electroactive biofilms (EABs), which grow on the anode under anaerobic conditions. Geobacter sulfurreducens is the most extensively studied electrogenic bacterial species because of its simplified electron transport mechanism and high activity [24,25]. The anode-adhered EABs catalyze the oxidation of organic molecules, producing electrons, as shown for acetate in Equation (1).
CH_3_COO^−^ + H_2_O → CO_2_ + 7H^+^ + 8e^−^(1)

After metabolic oxidation by *G. sulfurreducens* electrogens, electrons are externally conducted through the EAB external electron transport chain until they reach the anode via conductive pili and cytochrome enzymes. In other EAB types, electron transport may be accomplished or aided by electron-shuttling compounds. After injection into the anode, electrons enter an external electrical circuit that is connected to the cathode. The electrons flowing through the external circuit perform work according to the current and the load that it passes through it before finally being consumed in a reduction reaction at the cathode. There are many catholytes used in the literature, but as is the case in this work [26], ferricyanide is most often used during reactor development because its fast reduction kinetics from ferricyanide to ferrocyanide (Equation (2)) eliminate reaction bottlenecks at the cathode, allowing the performance of the anode-adhered EAB to be studied.
Fe(CN)_6_^3−^ + e^−^ → Fe(CN)_6_^4−^(2)

Typically, an ion transport membrane is placed between the anode and cathode chambers; this membrane selectively transports certain ions while maintaining the separation of the other solution constituents [27,28]. Typically, this is a proton exchange membrane that enables H^+^ produced in Equation (1) at the anode to participate in oxygen reduction reactions at the cathode in Equation (2). It should be mentioned that anion exchange membranes have recently been demonstrated to produce among the highest power densities recorded (approximately 9 W m^−2^) by admitting OH^−^ to the anode chamber to counteract EAB acidification [29].

After years of continuous MFC development, the emergence of small MFCs has been demonstrated to provide high power densities with diverse applications such as portable and even implantable energy sources [30,31], as well as others [32]. Among such small devices, microfluidic MFCs have many advantages, including laminar flow [33,34,35], reduced sample volume, and low cost. Recently, the problem of gas diffusion through a typical microfluidic fabrication material, polydimethyl siloxane, was solved [20]. Since then, our group has exploited highly predictable flow properties around EABs in microchannels and constant replenishment of reagents to minimize boundary layers [36], to counteract the common problem of power overshoot [37], and to optimize the operational conditions and protocol [38]. These works have culminated in demonstrations of among the highest power and current densities ever recorded in miniaturized MFCs [21,38]. Other authors have also been able to exploit microfluidic MFCs with simulations, adding deeper potential for optimization of outputs and comprehension of fundamental mechanisms [39,40]. 

While microfluidic channels benefit from high surface area/volume ratios, until now, microfluidic MFCs have generally fallen short of fully exploiting this, instead using just a fraction of the available channel wall space to mount electrodes. This limits coulombic efficiency (CE) because diffusion limits transport of nutrient molecules from reaching the EAB, which in turn limits current and power outputs.

In this study, a new four-electrode microfluidic MFC design was tested with the goal of maximizing the electroactive surface area within the device. This design introduces a second pair of electrodes on top of the channel, to complement and further the bioelectrical conversion of acetate on the bottom-side electrode. However, gravitational effects are known to affect biofilm growth [41]; these effects applied to EABs are addressed here for the first time. After overcoming the observed challenges in achieving uniform EAB performance on both anodes, we connected electrodes in different configurations and analyzed their power, internal resistance, and coulombic efficiencies at different EAB ages and flow rates. It is demonstrated that under optimal conditions, the four-electrode setup could double outputs compared to a two-electrode microfluidic MFC counterpart.

## 2. Methods and Experiments

### 2.1. Fabrication of a Four-Electrode Microfluidic MFC

There are a range of microfluidic fuel cells archetectures presented in the literature, with the most popular being a side-by-side electrode arrangement, whereby two electrodes are embedded on the same wall of the microchannel [42,43] such that co-flowing anolyte and catholyte streams contact each electrode individually. In this work, the four-electrode microfluidic microbial fuel cell (MFC) was constructed using 2 identical microchannel layers, each containing 2 embedded side-by-side graphite electrodes, as shown in Figure 1. This is the same process described previously [20], except that the 2 electrode-containing microfluidic layers are fabricated and bonded together instead of a single layer being sealed with a passive glass layer. To maintain the same channel height as in the previous work, both electrode-containing microfluidic layers were 80 µm thick, resulting in a channel height of 160 µm in total. To summarize the fabrication process in this work, each electrode-containing microfluidic layer was fabricated from polydimethylsiloxane (PDMS) (Sylgard184, Ellsworth adhesives, Germantown, WI, USA) with two embedded graphite electrodes in a side-by-side configuration, rather than the face-to-face configuration preferred by some authors [44]. This was accomplished by first placing the electrodes in contact with the channel feature of the photoresist mould and then casting PDMS against the assembly, so that the electrodes became embedded after solidification. This was followed by bonding the two layers together (Figure 1a,b). In this work, we used so-called bridge electrodes, which reduced the potential for leaking and enabled flexible placement of the electrode within the channel as described previously [38]. The device was designed to have the two electrodes on each electrode-containing microfluidic layer placed 4 mm apart (Figure 1c), resulting in an electrode/channel contact of 10 mm (downstream) by 4 mm (laterally). To avoid air intrusion, glass was bonded on the top and bottom of the assembled device, and epoxy was applied to all tubing and exposed PDMS surfaces as seen in Figure 1d.

A total of 5 microfluidic four-electrode MFCs were fabricated and used for this work.

### 2.2. Preparation of G. sulfurreducens Electrogenic Bacteria and Medium Solution

A nutrient solution was prepared by dissolving the following chemicals in 1 L of distilled water: 1.5 g NH_4_Cl, 0.6 g NaH_2_PO_4_, 0.1 g KCl, 2.5 g NaHCO_3_, 0.82 g CH_3_COONa (10 mM), 0.1 g L-cysteine hydrochloride, 10 mL ATCC^®^ MD-VS™ vitamin supplement, and 10 mL ATCC^®^ MD-TMS™ trace mineral supplement. A fumarate solution, used as the electron acceptor for electrogenic bacteria, was prepared by adding 1.6 g Na_2_C_4_H_2_O_4_ (40 mM) to 200 mL of the nutrient solution described above. Before adding the fumarate and vitamin/trace mineral supplements, the solution containing all the chemicals mentioned above was autoclaved at 110 °C and 20 PSI for 15 min. After it cooled down to room temperature, we added the sterilized vitamin/trace mineral supplements to the solution and added fumarate to 200 mL of the solution. The pH was adjusted to 7. The nutrient medium was then moved to an anaerobic glove box, and oxygen in solution was removed overnight. In addition, 50 mM potassium ferricyanide (K_3_[Fe(CN)_6_]) was prepared as the electron acceptor for the microfluidic MFC. It was dissolved in a 0.1 M phosphate-buffered saline (PBS) solution consisting of 75.4 mM sodium phosphate dibasic (Na_2_HPO_4_) and 24.6 mM sodium phosphate monobasic (NaH_2_PO_4_) in distilled water. All chemicals were obtained from Sigma-Aldrich (Oakville, ON, Canada), as used previously for the same nutrient solution recipe [20].

Frozen beads containing *G. sulfurreducens* (strain PCA, ATCC 51573, gifted by Derek Lovley) were removed from a −80 °C storage temperature and grown in a fumarate-containing nutrient medium. After 7–10 days in an anaerobic glove box, subcultures were ready for inoculation into the microfluidic MFC. This was conducted in a parallel co-flow, with ferricyanide as the catholyte, using two separate syringe pumps (New Era Pump Systems Inc., Farmingdale, NY 11735, USA): one for the anolyte and one for the catholyte. After 24 h, the inoculum was replaced by a sterile acetate source without any fumarate. The flow rate of acetate solution is defined as Q_Ac_, and that of ferricyanide is defined as Q_Fe_.

### 2.3. Electrochemical Measurements and Calculations

In this work we followed the established measurement and calculation protocol from leaders in the field, which is reviewed briefly here [45,46]. The anodes and cathodes were electrically connected through an external resistor box to close the circuit between them. According to Ohm’s law, the current (I) passing between the anode and cathode, through the external resistor (R_ext_), can be calculated by the potential (U) across the external resistor and the external resistor value (R_ext_) set on the resistor box:(3)$$I=\frac{U}{R_{ext}}$$
Using both I and U, the power (P) can be calculated as follows:P = UI(4)

A potentiostat (PARSTAT MC, Princeton Applied Research, Oak Ridge, TN, USA) was connected across the R_ext_ to conduct standard measurements, the most basic among them being the cell potential. It should be noted that this system did not have a reference electrode; therefore, the counter and reference leads to the potentiostat were both connected to the cathode, and the working lead was connected to the anode. This setup was also used to generate polarization curves (U versus I) by applying linear scan voltammetry (LSV) or by switching external resistor values (known as the constant-resistance technique). In order to limit artifacts in polarization and power density curves, the scan rates used were no higher than 2 mV s^−1^ [37,47,48,49,50]. Using Equation (4), power density curves were generated, and the maximum-power point was determined. The external resistance (and, equivalently, the current) that provides the maximum power is obtained when R_ext_ = R_int_ [45,46]. Therefore, the optimal R_int_ was obtained from Equation (3) given the current and voltage values at the maximum-power point.

Power and current were normalized (Equation (5)) to compare with other systems, a measure that is especially important when comparing results with the majority of the published works, which are conducted at size scales that are significantly larger.
(5)$$P_{A}=\frac{P}{N}$$
where N is a normalization constant that can be calculated in different ways to account for the system limitations. In some systems, where the anode area is not limiting (e.g., macro-systems using brush anodes), it has been shown that the membrane area between the anode and cathode compartments should be used for normalization [51]. Preliminary studies on microfluidic MFCs have shown that electrolyte contact between the anode and cathode compartments is limiting [21]; therefore, we normalize by the microchannel cross-section (height multiplied by length), using N = 1.6 mm^2^ in Equation (5), as performed previously [21,38].

The consumed concentration of acetate (Δ[Ac]) is calculated as follows (Equation (6)):(6)$$\Delta [Ac]=\frac{I}{nFQ_{Ac}}$$
where I is the raw current, $Q_{Ac}$ is the flow rate of acetate, F = 96,485 C mol^−1^, and n is the number of electrons produced per molecule of acetate that is oxidized. This is usually taken as 8, based on Equation (1); recently, however, it has been shown that at high [Ac] (e.g., >5 mM), n may be reduced by as much as 10% due to adjustments in bacterial metabolism to support biomass growth [52]. Unless otherwise stated, we use n = 8 in this work.

Periodic maintenance steps were required. These included sudden changes in measured chronoamperometric outputs due to syringe replacement, flow rate cycling for bubble removal, electrode reconnections, changing external resistors, and switching between electrochemical measurements. In such cases, we stopped data acquisition until the experiment is resumed, and we note this in relevant figure captions.

### 2.4. MFC Configurations

Different electrical configurations were tested for the four-electrode microfluidic MFC. First, we consider each pair of electrodes in separate layers in Figure 2a as different MFCs sharing the same channel (MFC_a_ and MFC_b_). By connecting both anode and cathode pairs in MFC_a_ and MFC_b_, we were also able to obtain a parallel-stacked MFC (MFC_a||b_) with one electrically connected anode and one cathode, as shown in Figure 2b.

### 2.5. SEM Imaging and Image Analysis

Scanning electron microscopy (SEM) was conducted to observe the morphology, thickness, and distribution of the biofilm attached to the surface of the anode. First, a fixation solution (2.5% glutaraldehyde (C_5_H_8_O_2_) in phosphate buffer) was allowed to flow through the channel at a flow rate of 0.5 mL h^−1^ for 2 h in a fume hood. Then, both anode electrodes were cut out from the device and soaked in the same fixation solution overnight. The next day, the electrodes were rinsed 3 times for 5 min each in 0.1 M cacodylate buffer (Na(CH_3_)_2_AsO_2_), then post-fixed in 1% osmium tetroxide (OsO_4_) for 90 min and washed again 3 times for 5 min each in cacodylate buffer. Subsequently, they were dehydrated in a graded ethanol series (30%, 50%, 70%, 90%, 95%, 100%; 10 min each), followed by two washes in 100% ethanol for 20 min. The electrodes were then dried with CO_2_ in a critical point dryer (Ladd Research Industries #2800, Essex Junction, VT, USA) for 45 min and mounted on aluminum stub (#75260, EMS) using aluminum-backed carbon tabs (#77828-08, EMS). Finally, the stubs were gold-sputtered using a SEM 950×/350 s sputter coater, and top-view SEM images were obtained (JEOL 6360LV, JEOL Ltd., Tokyo, Japan). Cross-section SEM images were collected next by plunging the electrodes into liquid N_2_ for 2 min, cracking with tweezers, and then immersing them in 100% ethanol to avoid moisture condensation on the specimens. The previous steps were repeated, starting with critical point drying, to observe the side view and obtain the thickness of the biofilm. Analysis of SEM images was used to obtain EAV thickness using software (ImageJ 1.54f, National Institutes of Health, Bethesda, MD, USA).

### 2.6. Computational Fluid Dynamics Simulations

A three-dimensional computation fluid dynamics model (COMSOL, Inc., Stockholm, Sweden) was used to monitor the conversion at the electrode surfaces in the 2- and 4-electode MFCs at 25 °C. The model included physics for electrochemical processes at the electrode, enabling generation of simulated polarization curves, and sensitivity analysis of polarization losses from causes including ohmic, mass transfer, and charge transfer limitations. The model consisted of three domains: (1) a solid anode-adhered EAB anode, (2) an electrolyte region, and (3) a solid cathode. A physics-controlled meshing technique was employed to optimize mesh geometry. We used a “finer” mesh that included approximately 2.7 × 10^6^ mesh elements and obtained a relative mesh error of 1.3 × 10^−3^%, converging to 0% error compared to the maximum mesh density that the software could provide (Table 1).

## 3. Results and Discussion

### 3.1. Growth of G. sulfurreducens in Four-Electrode MFC

*G. sulfurreducens* were inoculated into the system, with MFC_a_ and MFC_b_ being electrically separated via a connection to a separate external resistor (initially R_ext_ = 100 kΩ for both). This enabled separate monitoring of the growth and maturation process on the top (MFC_a_) and bottom (MFC_b_) sides (Figure 3a). We found that both MFCs matured to a stable voltage after 9 days; however, MFC_b_ (bottom side) produced significantly stronger voltage and current (Figure 3b). The better performance of the bottom MFC allowed the application of lower R_ext_ values, thus promoting further increases in current (Equation (3)) and power (Equation (4)).

The resistors were gradually reduced for both MFCs as the system matured, according to Table 2. At the end of the first growth analysis window (ca. 15 days), the bottom MFC (MFC_b_) produced significantly higher current (29.7 μA) than the top MFC (MFC_a_; 11.6 μA). We repeated the preliminary growth five times in separately fabricated four-electrode MFCs under the same conditions (flow rates and concentrations). In each case, the bottom MFC produced stronger output than the top MFC, leading to the hypothesis that gravity played a role in EAB growth efficiency. The reader is directed to the Appendix A, where polarization and power density curves from two such replicates are presented after 2–3 weeks of culture, all showing higher performance for the bottom MFC, independent of measurement conditions: flow rate or measurement technique (the constant-resistance technique or LSV). To investigate the hypothesis that gravity was the reason for the preferential growth on the bottom MFC and not, for example, asymmetries between the top and bottom sides’ fabrication or world-to-chip connections, we flipped the device after about 2 weeks, placing the original bottom-side MFC (MFC_b_) on the top side and vice-versa, and left the device to acclimate to the new conditions. Data after about 3 weeks of total growth time shows that the difference in current between the two MFCs had decreased, mainly due to the increase in current from MFC_a_ to nearly 20 μA (Figure 3c), with nearly no impact on MFC_b_. We electrically connected the two MFCs into the MFC_a||b_ (parallel) configuration and allowed the system to grow until approximately 2 months at an external resistance of R_ext_ = 10 kΩ (Figure 3c). After electrical separation, both currents were approximately the same (with R_ext_ = 10 kΩ for both MFC_a_ and MFC_b_), indicating that the culture under flipped conditions helped to normalize the performance of both MFCs. To investigate whether these improvements in the MFC_a_ were permanent or reversible, we flipped the device a second time. Long-term measurements of the individual MFCs outputs were stable over time, indicating that the system had previously reached full optimized maturity and that gravity no longer played a role (Figure 3d).

### 3.2. Power and Current Outputs at Different Ages

To assess whether the maximum power of the four-electrode MFC was significantly improved over that of the two-electrode MFCs, we collected LSV curves, after final maturity had been achieved. We collected polarization and power density curves of MFC_a_, MFC_b_, MFC_a||b_ after about 10 weeks. As seen from Figure 4, the power density values (P_A_ normalized by the channel cross-section) for MFC_a_ and MFC_b_ when electrically separated during LSV tests were nearly identical (P_A_ = 4 W m^−2^). This demonstrates the crucial role of inverting the MFC within the first 3 weeks of growth. Unsurprisingly, when the polarization and power density curves were collected for the MFC_a||b_ (using a potentiostat to conduct LSV), the measured maximum power density had nearly doubled to P_A_ = 7.09 W m^−2^. Based on the maximum-power points for each power density plot, the internal resistance (R_int_) values for MFC_a_ and MFC_b_, were each calculated to be approximately R_int_ = 15 kΩ, while R_int_ for MFC_a||b_ was nearly half this value.

Another objective for the four-electrode MFC was to increase the acetate conversion efficiency. We analyzed Figure 4 based on the maximum current produced. At a flow rate of 4 mL h^−1^ and R_ext_ values corresponding to the respective R_int_ values, the maximum current density was 8.49, 9.29, and 18.23 A m^−2^ for MFC_a_, MFC_b_, and MFC_a||b_, respectively. Using Equation (4), we calculated the change in acetate concentration as Δ[Ac] = 0.102 mM for MFC_a||b_, compared to half of that value for either MFC_a_ or MFC_b_ individually. While this only represents about a 1% change in concentration, this is significantly higher than reported in previous works using similar flow rates [20]. It should be noted that a significant overshoot in the Figure 4 data indicates that the measurement scan rate might have been too high for the MFCs to respond properly, resulting in artificially low currents on the high-current side of the power and polarization density curves [37]. We show the power density and polarization curves for lower flow rates (Q_Ac_ = 0.4 mL h^−1^, Q_Fe_ = 0.2 mL h^−1^) in the Appendix A.

### 3.3. Effect of Flow on MFC Performance

After demonstrating how to avoid disparities between the performance of top and bottom MFCs, resulting in a near doubling of current and power densities the four-electrode MFC (MFC_a||b_), we investigated the impact of experimental parameters on these outputs, notably the flow rate. It is well known that outputs can be further increased with flow rates; however, high flow rates also reduce hydraulic retention time in the MFC. This limits interaction time between the acetate solution and the anode-adhered EAB. Therefore, for high acetate conversion, the acetate flow rate (Q_Ac_) should be reduced, as predicted by Equation (6). We evaluated the performance of the mature system (>10 weeks) at flow rates ranging from 0.1 to 20 mL h^−1^. The resulting power densities are plotted in Figure 5a for MFC_a||b_ and individual electrode pairs comprising MFC_a_ and MFC_b_ (note the semi-log plot). The figure shows a rapid increase in P_A_ versus Q_Ac_ at low flow rates, and it became stable at high flow rates. On the other hand, when normalizing power and current by Q_Ac_, the opposite trends were observed. That is, the flow-rate-normalized power, previously referred to as volumetric consumption rate (VCR)-normalized power, P_VCR_ [21], or normalized energy recovery, NER [53], was increased to P_VCR_ = 7 mW L^−1^ d at Q_Ac_ = 0.1 mL h^−1^ for MFC_a||b_. The flow-rate-normalized current, is nothing more than Δ[Ac] [21], represented by Equation (6). We calculated Δ[Ac] and confirmed that it increased with the same trend as P_VCR_ when Q_Ac_ was reduced, reaching Δ[Ac] = 2.27 mM at Q_Ac_ = 0.1 mL h^−1^ for the MFC_a||b_. In other words, the conversion was 22.6%, or approximately twice that of individual electrode pairs (MFC_a_ and MFC_b_). These results are highlighted in Figure 5b. We note that recent studies show that the electron generation rate may be lower than eight per molecule of acetate at high concentrations due to a shift in the metabolism, which recycles about 10% of electrons from acetate oxidation for biomass synthesis [52]. If this were true in the present case, then the acetate conversion would be 25%. We also note that in the present study, the flow-rate ratio between the acetate anolyte and ferricyanide catholyte solutions was 2:1. This was required to avoid catholyte cross-over to the anode-adhered EAB, indicating a device malfunction (likely partial blockage in the downstream line of the catholyte side). Eliminating this problem would reduce the Q_Ac_ required for accurate co-flow interface between the two compartments, thereby increasing Δ[Ac].

### 3.4. Simulated Concentration Profiles

We conducted simulations to visualize the effect of the second pair of electrodes on the concentration profile in the microfluidic MFC_a||b_. Figure 6 shows the simulated concentration profile in the channel cross-section along a portion of the y-direction (colinear with the flow direction in the downstream direction) in the vicinity of the anode-adhered EAB(s). The thickness of the EABs was set at 20 µm based on SEM imaging as discussed in the next section. The simulation results in Figure 6a revealed that in certain regions of the four-electrode device (MFC_a||b_), acetate consumption was nearly five times higher than in Figure 6b for the two-electrode device (e.g., MFC_a_). Specifically, the expanded electrode surface area on both the top and bottom of the four-electrode device facilitates Ac capture by biofilms in two segregated locations, thereby increasing electron production.

### 3.5. SEM Imaging

Following the completion of the experiment, we measured the obtained SEM images to observe and measure the biofilm formation on the surface of the top and bottom anodes, as shown in Figure 7a,b. These images show tightly packed EABs with some fissures, though it should be noted that the fixation and dehydration processes may have changed the qualitative appearance of the EABs. Additionally, SEM cross-sections were obtained, which provided a measure of the EAB thickness. Two representative images are shown in Figure 7c,d, in which the EABs were measured to be 19 and 26 µm thick, respectively.

### 3.6. Discussion

This paper presents the results of a newly designed four-electrode microfluidic microbial fuel cell (MFC). The objective of this design was to enhance the conversion efficiency of acetate and improve power outputs by embedding two anodes and two cathodes on the top and bottom of the microchannel. Here, we investigate adding a top-side electrode pair rather than expanding the bottom-side electrode further down the length of the channel so that the electroactive surface area/volume ratio is increased, thus reducing the average diffusion length required for acetate molecules to encounter the EAB. This is superior to multiplying the length of the electrode against one wall, as it is known that progressive nutrient consumption along the length of a single electrode can result in depletion that can cause limitations such that the reaction rate becomes concentration-dependent (pseudo-first-order region of Michaelis–Menten kinetics) [54]. However, as this work is, to our knowledge, the first attempt to grow EABs on a downward-facing electrode, we first highlight the interesting discovery that gravity appears to significantly influence EAB maturation based on the repeatable discrepancy in growth rates between upward- and downward-facing anodes. This effect has been observed in non-electroactive biofilms under low laminar flow rates [41] and appears to have a correlation with observations that planktonic bacteria show less virulence and resistance to medication in microgravity situations [55,56]. We believe that this observation is the first one made in electroactive biofilms. Electroactive bacteria used as sensors for real-time respirometry can also prove useful for making generalized insights into biofilm growth on inverted surfaces. In any case, this effect can have important implications for space deployment of MFC systems. Furthermore, this overlooked point may actually be worse for terrestrial MFCs where negative gravitational fields are generated on downward-facing electrodes, as in our system. This appears not to be a major issue for brush electrodes, which show growth on the bristles’ top and bottom sides, likely due to the short distance between the two sides [57,58]. However, this subject should be investigated further in the case that large, unconnected downward-facing electrodes are used. For example, this should include a test of our hypothesis, based on literature reports in non-electroactive biofilms, that increased shear flow can compensate for growth inhibition in low or negative gravity [41]. Microfluidic studies are poised to contribute to such inquiries due to the ability to generate highly controlled shear forces on a chip. In the meantime, on Earth, one can oppose the negative effects on EAB growth rates on downward-facing electrodes by simply flipping the entire system during inoculation.

After overcoming gravitational effects, uniform behaviour was permanently achieved on both anode-adhered EABs, independent of their orientation after full maturation. In this state, the power output and acetate conversion of the parallel connected (MFC_a||b_) doubled to 7.1 W m^−2^ and 22%, respectively.

It is also worth noting that once the outputs began to normalize after flipping the device, the small fluctuations in the current became very similar (Figure 3b,c, and especially Figure 3d). The most likely explanation for this is the effect of imperfections in the flow rate, which, in microfluidics research, are well known to result in small disturbances in flow rate [59]. Since the current and power outputs of an MFC are directly related to the flow rate, some of what appears to be noise in the current is actually the result of small pulsations in the flow rate that affect both electrode pairs (MFC_a_ and MFC_b_). We also do not exclude the possibility that after the EAB reaches sufficient thickness, some portions of the biofilm have made contact between the MFC_a_ and MFC_b_, thereby partially coupling their outputs. There is an enhanced possibility of this occurring at the vertical side wall, where flow is slowest and where we often exploit the continuous biofilm growth spanning the entire vertical distance of the microchannel for calibration in microscope-based optical density measurements [60]. The potential of such a top-to-bottom bridging material to effectively transform the entire microchannel interior to electroactive surface area merits future investigation.

While we successfully proved the concept of a four-electrode MFC, we note that there is room for improvement. For example, it should be noted that the two facing anode-adhered EABs will result in a constrained liquid volume, which can result in higher shear stresses and a corresponding loss of biomass. In a previous study featuring a two-electrode MFC that otherwise had similar dimensions to those used here [61], we verified that *G. sulfurreducens* cells were present in solution and exploited this to extract and re-culture them. Future studies should investigate whether enhanced shear stresses in face-to-face anode-adhered EABs result in erosion based on measurements of effluent cell density, especially in light of previous studies that have shown a complicated relationship between the circulating (planktonic) cell density and shear stress in non-EABs [62]. We sampled a range of literature reports on acetate-fed microfluidic MFCs and plotted the conversion percentage based on reported current outputs and flow rates (Figure 8) [21,22,37,63,64,65,66,67,68,69]. From this figure, we can see that the output obtained with the current four-electrode system was nearly the highest in the literature. Examining some of the better-performing systems can give guidance on future implementations of a four-electrode MFC. Previously, we demonstrated major limitations of the current design, which relate to the width of the microchannel. At positions near the far electrode edges where the electric field is diminished, acetate consumption becomes rapidly reduced [21]. Thus, while the four-electrode MFC can double the current and power outputs, reducing the electrode and channel widths is key to improving overall conversion (see Appendix A for more details, including simulation results confirming that for the present situation). For example, previous work in our group has focused on reducing the electrode width to eliminate portions of the anode that are far from the cathode [21]. In the same work, we showed that optimizations in the channel structure could stabilize the co-flow interface, which, together, resulted in a two-electrode MFC that generated the highest power density at the time [21]. A stable co-flow interface could also help overcome the present problem whereby we were forced to use Q_Ac_ > Q_Fe_ to avoid ferricyanide cross-over to the anode side, which reduced the Δ[Ac] (Equation (6)). In another recent paper with a two-electrode MFC, electrode placement away from the channel wall (eliminating contact with the low-flow-velocity anolyte solution) resulted in an MFC with the highest recorded current density of 63 A m^−2^ [38], or about nine times higher than we obtained for MFC_b_ when subjected to a similar flow rate of Q_Ac_ = 4 mL h^−1^ (Figure 4). In yet another paper featuring a two-electrode design with electrode separation of less than 1 mm, acetate conversion was over 30%. It stands to reason that outputs from each of these examples could be doubled using a four-electrode MFC, and combining the critical innovations in each could provide a synergistic effect. We note that membrane microfluidic MFCs typically use anodes and cathodes in a face-to-face orientation, leaving no room for a second pair of facing electrodes, as shown for the side-by-side electrode orientation used here. In such a case, exceptional conversion efficiency has been noted at very low flow rates, for example, more than 65% by Choi et al. [62]. However, the lack of a membrane and the ability to place anodes and cathodes close together in side-by-side MFCs may make it possible to surpass this benchmark in future systems.

## 4. Conclusions

A comparison of two- and four-electrode microbial fuel cell configurations is presented. In the four-electrode design, individual electrode pairs (forming individual MFCs) are placed on the top and the bottom of the main flow channel. It was discovered that gravity plays a critical role in the efficacy of inoculation and initial EAB growth. This effect negatively impacts the (top-side) downward-facing electrodes. Apart from having implications for microgravity applications, such as in MFC deployment in space, this can be important for terrestrial MFC applications, particularly for downward-facing electrodes, which experience a negative gravitational field compared to upward-facing electrodes. A methodology for permanently overcoming this problem resulted in normalized outputs from top- and bottom-side MFCs as well as power outputs and acetate conversion rates that were doubled when they were connected in parallel (i.e., a single four-electrode MFC), reaching 7.1 W m^−2^ and 22% conversion. These very high values were obtained without any special adjustments to the channel geometry or flow considerations, both of which could further enhance outputs in future work.

## Figures and Tables

**Figure 1 micromachines-15-00961-f001:**
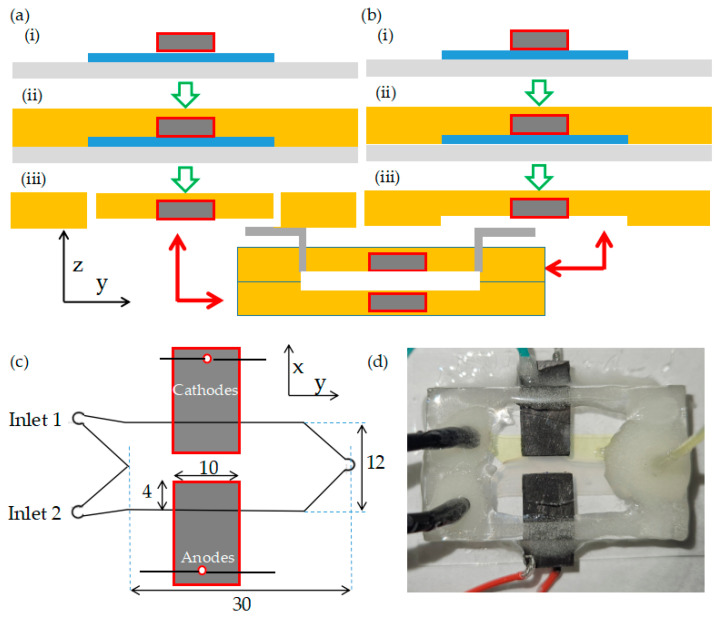
Fabrication of a double-sided microfluidic MFC with 4 embedded electrodes as viewed in the x-y plane (top view). (**a**,**b**) Fabrication of both electrode-containing microchannel layers with two embedded graphite electrodes: (i) Two graphite electrodes (dark grey) placed face down on a photolithographic microchannel mould (blue). (ii) A PDMS and cross-linker solution (yellow) poured into the mould. (iii) Access holes punched in the PDMS microchannel layer in (**a**). Schematic (**c**) and photo (**d**) of final device (with all dimensions in millimeters). Green hollow arrows show the progression of the fabrication for each layer, red arrow show the combination of each finished layer into the final device.

**Figure 2 micromachines-15-00961-f002:**
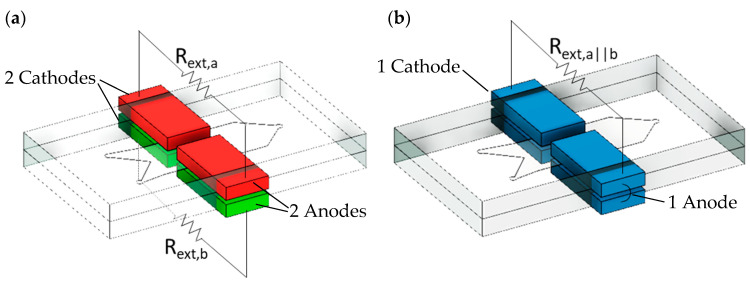
(**a**) MFC_a_ (red) and MFC_b_ (green) consisting of 2 side-by-side electrodes each at the top and bottom of the channel, respectively. Each separate electrode pair is connected to its own external resistor (R_ext,a_ and R_ext,b_). (**b**) Parallel connection of MFC_a_ and MFC_b_ into a single four-electrode MFC (MFC_a||b_, blue) with one electrically connected anode and cathode.

**Figure 3 micromachines-15-00961-f003:**
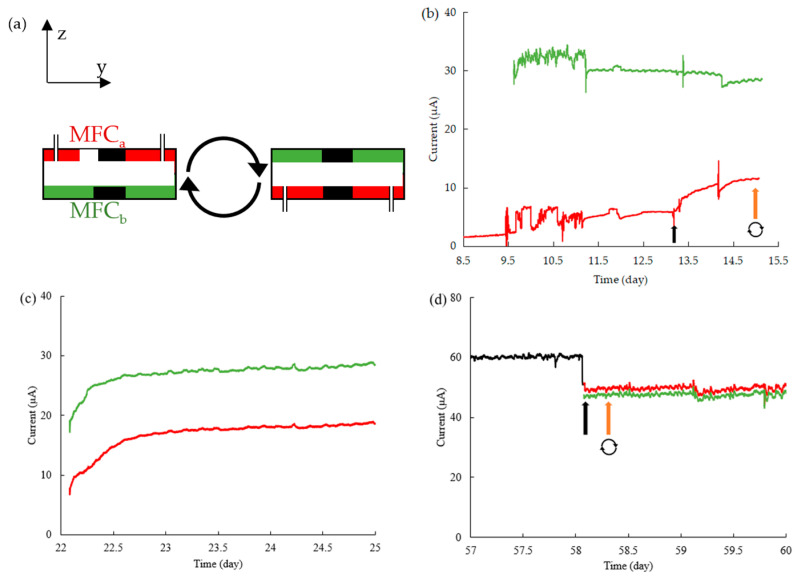
(**a**) Schematic for MFC orientation, with MFC_a_ (red) initially being on the top and MFC_b_ (green) initially being on the bottom. The white space between MFC_a_ and MFC_b_ represents the channel space, and the black rectangles represent electrodes. Note that, in the z–y plane, only 2 of the 4 electrodes are visible. Inlet and outlet tubing is connected through the MFC_a_ layer. (**b**) Growth curves with MFC_a_ (red) and MFC_b_ (green) starting with external resistances of R_ext_ = 100 kΩ, with MFC_a_ being switched to 50 kΩ at 13 days (black arrow). Rotating arrows near the 14-day mark the time that the device is flipped as shown in (**a**). The data acquisition was stopped for approximately 20 min at 11 days and 14 days due to syringe changes in MFC_b_ and MFC_a_, respectively. (**c**) Mid-stage mature phase after initial flipping in (**a**). External resistances are R_ext_ = 30 kΩ for the MFC_a_ and R_ext_ = 20 kΩ for the MFC_b_. (**d**) Current outputs from MFC_a||b_ (black) to MFC_a_ (red) and MFC_b_ (green) are electrically separated (black arrow) and flipped back to the original orientation (orange arrow). At all stages, the flow rates for acetate and ferricyanide were Q_Ac_ = 0.4 mL h^−1^ and Q_Fe_ = 0.2 mL h^−1^, respectively.

**Figure 4 micromachines-15-00961-f004:**
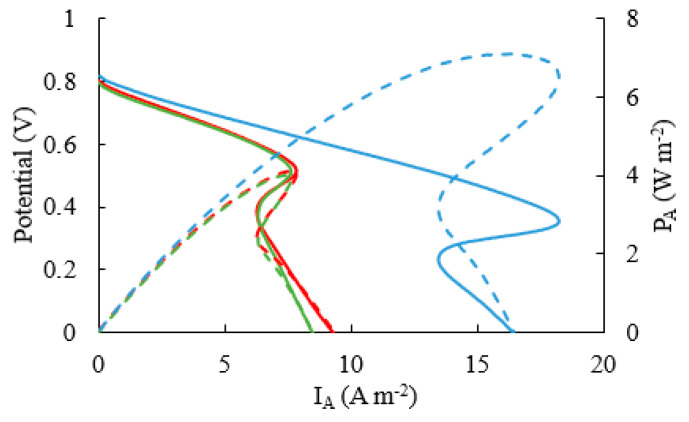
Polarization curves (solid line) and power density (dashed line) of MFC_a_ (green), MFC_b_ (red) and MFC_a||b_ (blue) at biofilm age of 68 days (Q_Ac_ = 4 mL h^−1^, Q_Fe_ = 2 mL h^−1^, scan rate 2 mV s^−1^).

**Figure 5 micromachines-15-00961-f005:**
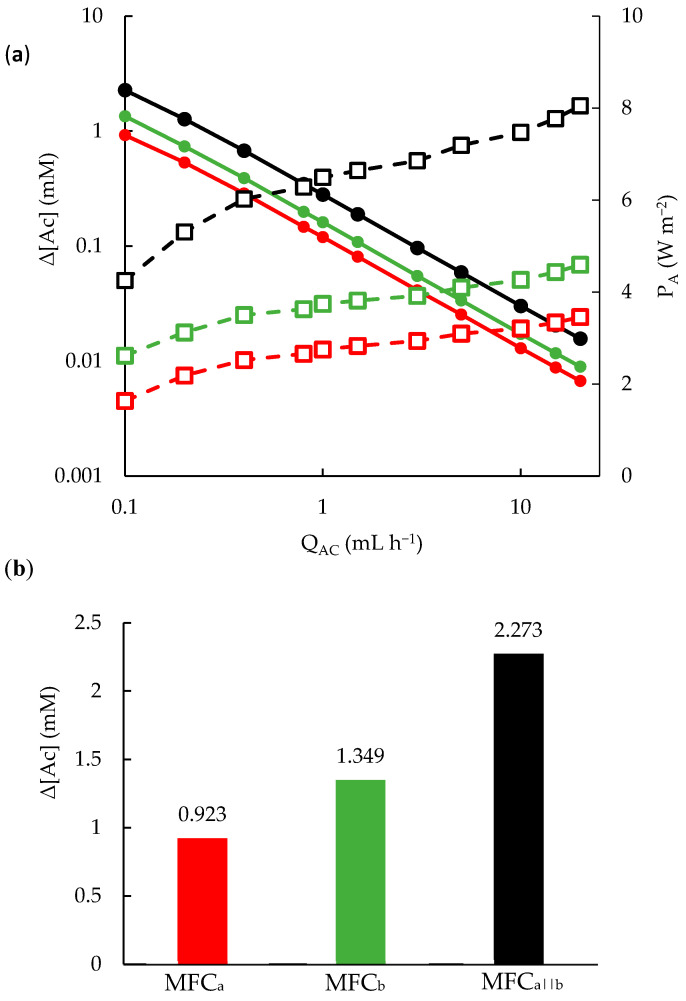
(**a**) Changes in acetate concentration (Δ[Ac]; solid circles) and power density (P_A_; open squares) with the flow rate of acetate for MFC_a_ at R_ext_ = d 20 kΩ (green), MFC_b_ at R_ext_ = 15 kΩ (red), and MFC_a||b_ at R_ext_ = 15 kΩ (black). (**b**) The calculated Δ[Ac] at Q_Ac_ = 0.1 mL h^−1^ for MFC_a_, MFC_b_, MFC_a||b_.

**Figure 6 micromachines-15-00961-f006:**
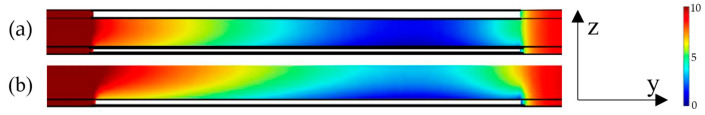
Simulation of acetate consumption in the anode compartment showing a preference for consumption near the edge closest to the cathode in (**a**) a four-electrode MFC (MFC_a||b_) and (**b**) a two-electrode device (e.g., MFC_b_ after 2 months). Both are cross-sections in the x-z plane. The concentration color bar is in units of millimolar.

**Figure 7 micromachines-15-00961-f007:**
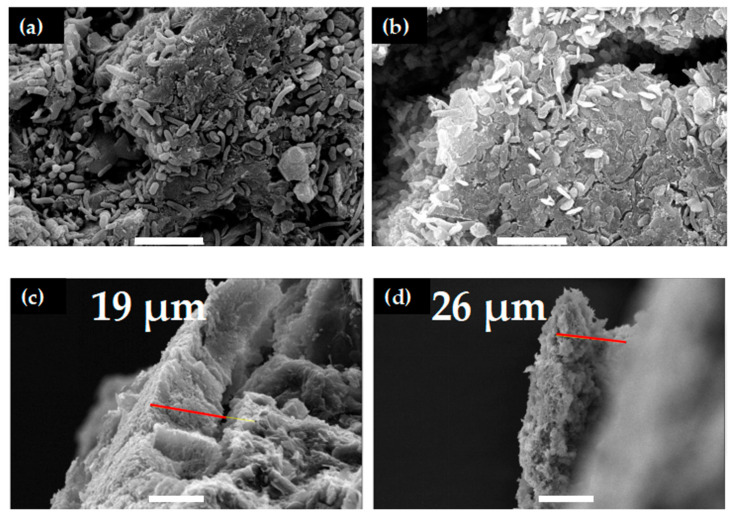
SEM images of the biofilm on the surface of the top anode (**a**) and the bottom anode (**b**). Side view of the biofilm in the middle of the top anode (**c**) and the bottom anode (**d**). With measurements of thickness along red lines. Magnification is 5000 times, and scale bars represent 5 µm in (**a**,**b**). Magnification/scales bars in (**c**,**d**) are 2000 times/10 µm and 1000 times/20 µm, respectively. All data were acquired using 15 kV.

**Figure 8 micromachines-15-00961-f008:**
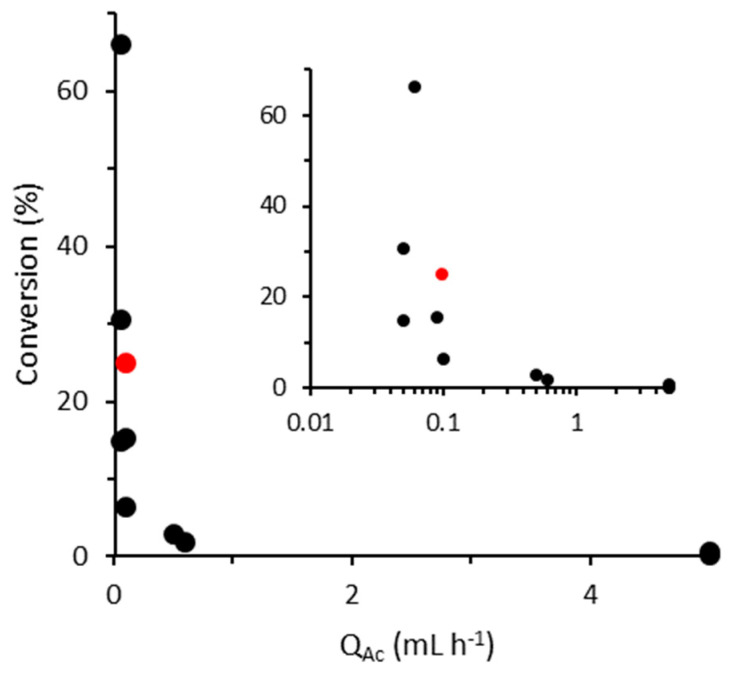
Literature values (black) and present value (red) of acetate conversion by a microfluidic MFC as a function of acetate flow rate (Q_Ac_) based on reported currents. For display purposes, the inset is plotted on a semi-log scale, with axis titles being shared with the main figure.

**Table 1 micromachines-15-00961-t001:** Summary of mesh independence analysis results.

**Pre-Set Mesh Density**	**Number of Mesh Elements**	**Relative Error (%)**	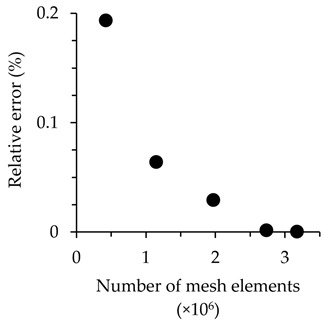
Coarse	421,983	0.19379	
Normal	1,142,143	0.06420	
Fine	1,964,457	0.02939	
Finer	2,738,854	0.00130	
Extra fine	3,173,758	0	

**Table 2 micromachines-15-00961-t002:** R_ext_ of MFC_a_ and MFC_b_ at different growth ages.

R_ext_ (kΩ)	MFC_a_	MFC_b_
Figure 3a	100, 50	20
Figure 3b	30	20
Figure 3c	10	10

## Data Availability

The original contributions presented in the study are included in the article, further inquiries can be directed to the corresponding author.

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
