# Peer review of "Defying Gravity to Enhance Power Output and Conversion Efficiency in a Vertically Oriented Four-Electrode Microfluidic Microbial Fuel Cell"

_micromachines, 2024, doi:10.3390/mi15080961_

Round 1

Reviewer 1 Report

Comments and Suggestions for Authors

In this work, the authors developed 4-electrode laminar flow microbial fuel cells (MFCs) with anode and cathode pairs positioned on top and bottom of the flow channel. The authors also developed a method to counter the slow biofilm formation on the top anode by inverting the MFC during the start-up period. Although the experiment and results are solid, the manuscript requires major improvements in English expression to ensure clarity for the readers. I suggest the authors revise the manuscript to clarify the English expressions and address the following comments before publication.

1.     Please reconsider the title. Using “but” here is a bit strange, maybe consider changing it to “Enhanced Power Output and Conversion Efficiency in a 4-Electrode Microfluidic Microbial Fuel Cell: Addressing Gravitational Challenges” or something similar.

2.     Page 2 line 39. Electron transfer by shuttling compounds is also a major pathway, please add it alongside conductive pili and direct contact.

3.     Page 2 line 41. Not sure what “do work” means here, could remove it for clarity.

4.     Page 13, Figure 6. Please include the colored scale/legend similar to that in Figure S3.

5.     Supporting information, page 1. In the legend for S1, “young MFCs” is also a strange expression, consider changing it to “starting-up MFCs” or something similar.

Comments on the Quality of English Language

The English expression in the manuscript is not clear. I have commented on the major issues, but I advise the authors to revise the manuscript for clarity. Having another person to proofread it may be beneficial.

Author Response

Thank you for your comments. We have addressed everything in the attached document.

Reviewer 2 Report

Comments and Suggestions for Authors

Presented paper discuss microfluidic microbial biofuel cell. Authors showing that 4-electrode approach can double power output and also indicate that gravitation effects have its impact on the power output. Paper is well written and present results gives interesting information for further development of microbial fuel cells. There are only several comments that should be taken into account before paper can be published:

-          Effect of gravitation on microbial biofilm formation is known and discussed in number of papers; author mentioned this in discussion section. It is better to discuss this effect in brief in introduction, to clarify goals of the research;

-          As it known for many publications electrodes area also influence to the power output. Does adding additional pair of electrodes can more increase power output in comparison with doubling electrodes surface area? Answer is not clear shown in the paper;

-          Section 2.1 – please consider adding simple schematic 3D picture of microfluidic MFC. This simplify presentation of the research for reader;

-          How many microfluidic MFC was made for this researches?

-          Line 107 - nutrient solution is your development or taken from literature for this strain? In second case citation is needed;

-          Line 123 – please add information concerning collection or institute who gives you this G.sulfurreducens strain.

-          Figure 3b – What cause current drop in 11 day?

-          Figure 3d – Current behavior looks very similar for both MFC after they was electrical separated, could you please discuss this effect?

-          Does any G.sulfurreducens bacterial cells were observed in the nutrient medium?

Author Response

Thank you for your comments. We have responded to everything in the attached document.

Round 2

Reviewer 1 Report

Comments and Suggestions for Authors

The revised manuscript has addressed all the issues I brought up.

Reviewer 2 Report

Comments and Suggestions for Authors

This paper can be accepted in the present form